# Tessellated Neural Networks: A Robust Defence against Adversarial Attacks

## Abstract

Data-driven deep learning approaches for image classification are prone to adversarial attacks. An adversarial image which is sufficiently close (visually indistinguishable) from a true image of its representative class can often be misclassified to be a member of a different class. It is possible for attackers to exploit the high dimensionality of image representations, as learned by the neural models, to identify adversarial perturbations. To mitigate this problem, we propose a novel divide-and-conquer based approach of tessellating a base network architecture (e.g., a ResNet used in our experiments). The tessellated network learns the parameterized representations of each non-overlapping sub-region or tiles within an image, independently, and then learns how to combine these representations to finally estimate the class of the input image. We investigate two different modes of tessellation, namely *periodic*, comprised of regular square-shaped tiles, and *aperiodic*, comprised of rectangles of different dimensions. Experiments demonstrate that the tessellated extension of two standard deep neural models leads to a better defence against a number of standard adversarial attacks. We observed that the decrease in post-attack accuracy values relative to the accuracy of the uncompromised networks is smaller for our proposed tessellated approach.

## 1 Introduction

Deep neural networks are known to be susceptible to adversarial attacks. Image representations learned by a deep neural network differ from their visual interpretation. Attackers exploit this fact by introducing imperceptible evasive perturbation in test images such that the victim network misclassifies them (Goodfellow et al., 2018; Machado et al., 2021). Defending neural networks against such adversarial attacks is of significant theoretical and practical importance.

Well known evasive attacks include the gradient based input perturbation strategies such as fast gradient sign method (FGSM) (Goodfellow et al., 2015), and the projected gradient descent (PGD) (Madry et al., 2018) methodologies. Non-gradient based attacks use norm bounded perturbations that change class membership through an optimization process (Andriushchenko et al., 2019). Universal attacks that are image-agnostic and add the same perturbation for all input images while still modifying the class labels are also prevalent (Moosavi-Dezfooli et al., 2017). Norm based attacks seeking to optimize the magnitude of perturbation in input images were subsequently proposed to victimize newer defence strategies (Carlini & Wagner, 2017; Croce & Hein, 2019). Patch attacks, which involve perturbing image segments rather than the image pixels, have also been attempted (Sharif et al., 2016; Yang et al., 2020). More recent attacks approaches include the use of ensemble-based strategies with a capability to adapt on the defence mechanisms employed (Tramèr et al., 2020).

As newer attacks are being proposed by researchers, developing models that are robust to adversarial attacks has attracted significant attention of the research community (Machado et al., 2021). Early defence strategies include adversarial training (Madry et al., 2018; Goodfellow et al., 2015) where a classifier is trained using both legitimate and adversarial examples to improve its robustness. Adversarial training restricts the defence only to the specific attack strategy using which the examples were generated. Other proactive defences retrain deep networks using the smoothed output probabilities over the class labels using the principles of network distillation (Papernot et al., 2016). Both these

retraining methods modify the gradient of the networks so as to allow fewer directions (subspaces) towards which the attacker might perturb the input image.

Input transformation is another popular defence strategy. In this approach the corrupted inputs are either detected and rejected before classification (Chen et al., 2017), or a preprocessing is performed to mitigate its adversarial effects. Various preprocessing strategies have been suggested towards this end. Adding a random resizing and padding layer in early part of the architecture (Xie et al., 2017), blockwise image transformation (AprilPyone & Kiya, 2020), cropping and rescaling (Guo et al., 2018), include some of these techniques. As a different thread of work, transformation of the features at the output of the convolution layers such as activation pruning (Goodfellow, 2018) and denoising (Dhillon et al., 2018; Liao et al., 2018) are often equally effective as defence mechanism. Input dimensionality reduction approaches based on PCA (Hendrycks et al., 2019), and spatial smoothing (Xu et al., 2017) are also found to provide robustness against attacks.

Besides adversarial retraining and input transformation, various other techniques have also been attempted as defence strategies in deep neural networks. Ensemble of classifiers are found to be more robust towards adversarial attacks (Tramèr et al., 2017). Data augmentation using GAN, generative models, and ensembles (Wang et al., 2021) has been widely studied in this context. State-of-art defences as reported in the RobustBench (Croce et al., 2020) benchmark dataset include those based on data augmentation for adversarial training (Rebuffi et al., 2021a), as well as those that are based on transformation or randomization of model parameters (Gowal et al., 2021).

Different from the input or feature transformation based approaches, architectural changes to a network topology is a promising means of achieving adversarial robustness (Huang et al., 2021). For instance, Du et al. (2021) and Pang et al. (2019) generate diverse structured networks for ensuring robustness against adversarial threats. An alternative convolutional network (CNN) architecture which randomly masks parts of the feature maps also demonstrates adversarial robustness (Luo et al., 2020). An advantage of this approach over its transformation-based counterparts is that it provides an effective defense mechanism that is mostly agnostic to attack strategies.

As a motivation of the work in this paper, we hypothesize that modification of the network structure leads to implicit feature transformation, cropping and masking. This, in turn, potentially results in improved robustness against adversarial attacks. Moreover, incorporation of diversity in network topology likely disrupts the gradients, and acts as an effective defence against ensemble attacks. Consequently, reconfiguring the topology of a network may provide effective defence against adaptive adversarial attacks. Since attackers exploit high dimensionality of image inputs to identify directions for adversarial perturbations (Machado et al., 2021), a *divide and conquer* strategy of processing smaller blocks of an input image, which is the basis of our proposed method in this paper, potentially restricts the exploitable space of the attacker.

**Our Contributions.** In this paper, we propose a tessellated deep neural network architecture - 'split and merge' based workflow that provides an effective defence mechanism against adversarial attacks. In our proposed approach, an input image is partitioned into blocks (tiles) according to a tessellation (tiling) pattern. Each region of the input image makes use of a separate branch in the computation graph to propagate its effects forward in the form of feature representations. The individual feature representations then interact with each other for the eventual prediction of an image class (see Figure 1 for a schematic representation). We investigate the use of two types of rectangular tessellation patterns, namely, regular grid tiling and tiling with non-uniform rectangles. As base networks (on which the tessellation is applied), we investigate standard deep networks for image classification, namely the ResNet50 (He et al., 2016), and show that the proposed approach turns out to resist standard adversarial attacks more effectively than a number of baseline defence methodologies.

## 2 Tessellated Neural Network

In this section, we describe our proposed method of tessellated neural architecture.

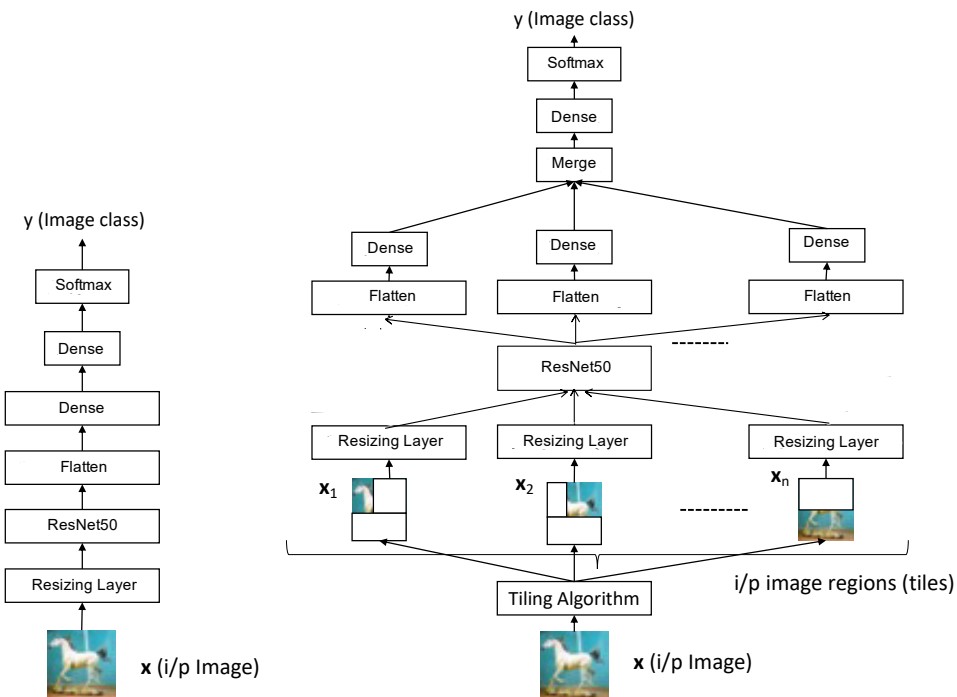

Figure 1: Architectural overview of the proposed approach of a tessellated neural network. The proposed method consists of partitioning an input image into several non-overlapping regions or tiles. We then make several copies of a base network, one for each of these tiles, each involving a separate set of learnable parameters. For simplicity, we illustrate our approach schematically on a resnet50 architecture (He et al., 2016) as the base network (as shown on the left). The tessellated version of the base network is shown on the right of the figure.

## 2.1 ARCHITECTURE OVERVIEW

The process of tessellating acts on a base neural network with a specific architecture. We use ResNet50 as base architectures and illustrate our idea schematically on it. Figure 1 presents the schematic overview of a tessellated network. Our proposed approach comprises of a split phase and a merge phase, which we now describe below.

**Split phase.** An input image is first partitioned into non-overlapping rectangular tiles using a tessellation scheme (more details in Section 2.2). and then the spatial resolution of the image is increased using the up-sampling layer and resize the image to 224X224.We then pass the output to the base network, one for each partitioned region or tile within the input image. Each of these tile consists of shared learnable parameters of the base network.

**Merge phase.** To enable an end-to-end training for the entire image, the outputs from the base network, each corresponding to a separate region of the image, are then merged together and passed through a subsequent feed-forward layer.

While the split phase enables learning an abstract representation of each different regions of an image dependently, the merge phase allows provision for an adjustment to these individual representations as per the downstream task objective.

It might be argued that the split phase incorporates some amount of information loss due to the fact that each time the base network is trained only with partial information. Consequently, the downstream task (e.g., that of image classification) effectiveness of a tessellated network is expected to be slightly lower than that of the base network itself (which, in contrast, leverages information from an entire image).

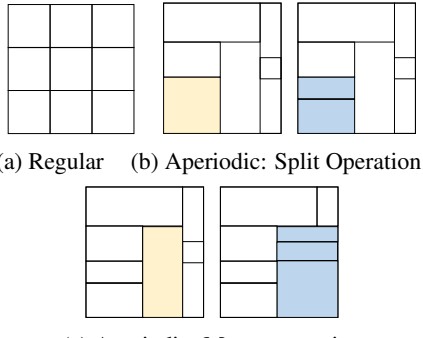

(a) Regular     (b) Aperiodic: Split Operation

(c) Aperiodic: Merge operation

Figure 2: (a) Regular tiling of a $9 \times 9$ image with nine $3 \times 3$ square tiles; (b) Illustration of the effect of the *split* operation on a tiling - the yellow colored rectangle on the left part of the figure is transformed into two blue rectangles shown in the tiling shown along the right; (c) Illustration of the effect of the *merge* operation on a tiling - the yellow colored rectangle on the left part of the figure is merged with the rectangles right-adjacent to it; this operation for this example does not increase or decrease the total number of tiles (as seen on the right part of the figure);

However, this is precisely the reason why such a tessellated network should provide a robust defence mechanism against adversarial attacks. This is because the gradients in the lower layers of a tessellated network are associated only with partial information from the input.

## 2.2 TILING METHODS

In this section, we describe two ways of partitioning an input image into non-overlapping regions. At this point, we would like to emphasize that the goal of the paper is to demonstrate that a tessellated neural network potentially provides an effective defence mechanism. Investigating other more involved ways of partitioning an input image, e.g., by overlapping regions, is left as a future work.

### 2.2.1 REGULAR TESSELLATION

The simplest tessellation that we consider is the uniform one, where each tile is a square (Figure 2). The parameter $k$ for regular rectangular tessellation denotes the number of squares used to cover $\vec{x} \in \mathbb{R}^{d \times d1}$, and is a perfect square, i.e., $k = m^2$ for some $m \in \mathbb{Z}$. To ensure regular tessellations in situations where $d$ is not a perfect square, we apply zero padding to those regions of the image not covered by the tiles. In the general case, partitioning a $d \times d$ square with $k = m^2$ tiles, leaves a residual region of area $2d(d-k) - (d-k)^2$ for zero padding. For instance, the residual region of a $28 \times 28$ image with 25 tiles is 159.

### 2.2.2 APERIODIC TESSELLATION

In aperiodic or non-uniform tessellation, an input image of size $d \times d$ is split into rectangular blocks of varying sizes, with a low likelihood that any two rectangles will be congruent, i.e., of identical area.

The basic idea to obtain an aperiodic tiling is to iteratively apply two operations, namely the *split* and *merge*. While applying the split operation each time to a current state (set of tiles) leads to increasing the number of tiles by 1, the application of merge operation, on the other hand, mostly reduces the number of tiles by 1. More details on the aperiodic tiling algorithm follows next.

We start with the initial state representing an entire image of size $d \times d$. We then successively apply the split operation until the number of tiles is $k'$ where $k' = 2k$. At this phase, we apply the merge operation successively until the number of tiles reaches from $2k$ to $k$. Both the split and the merge operations are applied on a randomly chosen rectangle from the set of tiles existing in the current state.

---

[1]Extending the argument to the more general case of rectangular images is straightforward.

| Dataset | Images | Dimension | classes |
|---------|--------|-----------|---------|
| CIFAR-10 | $60,000$ | $32 \times 32$ | 10 |
| CIFAR-100 | $60,000$ | $32 \times 32$ | 100 |
| PlantVillage | $43,447$ | $224 \times 224$ | 38 |

Table 1: Summary of the datasets used in our experiments.

In general, for the split operation, given a state comprised of $k'(0 < k' \leq 2k)$ states, we first randomly select one of the $k'$ rectangles and split it into two parts either horizontally or vertically. The position of the splitting line and its direction (horizontal or vertical) is chosen randomly, e.g. the blue-shaded rectangle in Figure 2b is split into two parts by the bold-faced line. In general, a rectangle of width $w$ and height $h$ is split into a pair of rectangles of dimensions $w \times p$ and $w \times h - p$ if the splitting direction is horizontal, or into the pair $p \times h$ and $w - p \times h$ if the direction is vertical with the condition that $p$, $w - p$ $h - p$ are all positive integers.

Similarly, for the merge operation, we first select a rectangle at random. We then merge it with other rectangles that are adjacent to it with respect to a direction (one of top, right, bottom or left). Figure 2 illustrates an example of merging a tile with the ones that are right-adjacent to it. The merge operation mostly leads to decreasing the number of tiles (although in some situations, the number of tiles after the operation remains the same as, as shown in Figure 2c).

As a sanity check to ensure not too small or too large tiles (input image regions in the tessellated network), we undo a split or a merge operation if it leads to creating a rectangle whose dimension is less than $5^2$ or higher than $(\frac{3}{4}d)^2$.

While it is, in principle, possible to generate non-uniform tessellations with other policies as well, e.g., with the use of Bayesian non-parametric space partition methods as surveyed in (Fan et al., 2021), we leave this as a part of future work, since estimating the optimal tiling strategy for a tessellated network is not the core objective of the paper.

**Why should tessellated networks be robust against adversarial attacks.** A tessellated network dedicates a separate computational branch for each of the tiles as shown in Figure 1, which are later combined via a feed-forward layer. This offers an implicit ensemble of the features computed for each region of an input image by each computational branch. Empirically, ensemble of diverse network structures have been shown to exhibit robustness (Du et al., 2021; Pang et al., 2019). Moreover, adopting a non-regular (and non-congruent) image tessellation scheme leads to diversity in the computational graph of the branches of a tessellated network.

On the other hand, non-regular tessellation patterns with large variation in sizes of the tiles may result in very large or too small sub-rectangles. Through our experiments, we found that this decreases the prediction effectiveness of the tessellated network on non-adversarial samples. This trade-off between performance on clean images and robustness to adversarial attacks has also been noted in previous studies. Generating non-uniform tessellations (those obtained with the split-merge operations) lead to better handling of this trade-off.

## 3 EXPERIMENTAL SETUP

### 3.1 DATASETS

Our main experiments were conducted on two standard benchmark image classification datasets. The first dataset that we use is the CIFAR-10 dataset, which consists of 60000 $32 \times 32$ colour images in 10 classes, with $60,00$ images per class. To investigate if the defence mechanism shows a consistent behaviour for more fine-grained (and thus a higher number of) classes, we use the CIFAR-100 dataset, for which there is a 10-fold decrease in the number of training instances available per class as compared to CIFAR-10. Moreover, in order to study the effectiveness of the proposed tessellated architecture on larger sized images, we evaluate the proposed approach on the 'PlantVillage' dataset consisting of $43,447$ $224 \times 224$ colour images in 38 classes. The dataset consists of images of diseased leaves from multiple crops Hughes & Salathé (2015). A summary of the dataset characteristics is presented in Table 1. In our experiment setup, we use data augmentation for classi-

Table 2: Adversarial robustness of Tessellated Resnet-50 (RN) with a perturbation of $\ell_\infty, \varepsilon = 8/255$, reported with optimal hyper-parameter settings, namely $k = 16$ for regular tiling (Reg) and $k = 20$ for aperiodic tiling (Aprd), where $k$ denotes the number of tiles in a tessellation. The smallest drops in accuracy and weighted accuracy values are shown bold-faced for each attack on the respective datasets.

| Attack | Metric | CIFAR-10 | | | CIFAR-100 | | |
|---|---|---|---|---|---|---|---|
| | | RN | TRN | | RN | TRN | |
| | | | Reg | Aprd | | Reg | Aprd |
| NONE | acc | 0.9417 | 0.8559 | 0.8260 | 0.7741 | 0.5174 | 0.5085 |
| | $acc_w$ | 0.9510 | 0.8939 | 0.8802 | 0.8186 | 0.6790 | 0.6721 |
| FGSM | acc | 0.4586 | 0.3914 | 0.3714 | 0.3207 | 0.1297 | 0.1363 |
| | $\Delta acc$ | **0.5130** | 0.5427 | 0.5607 | **0.5857** | 0.7493 | 0.7320 |
| | $acc_w$ | 0.5315 | 0.5421 | 0.4974 | 0.4583 | 0.3773 | 0.4151 |
| | $\Delta acc_w$ | 0.4411 | **0.3936** | 0.4349 | 0.4402 | 0.4443 | **0.3825** |
| PGD | acc | 0.1427 | 0.3574 | 0.3694 | 0.2044 | 0.2459 | 0.2608 |
| | $\Delta acc$ | 0.8485 | 0.5824 | **0.5528** | 0.7360 | 0.5247 | **0.4871** |
| | $acc_w$ | 0.1628 | 0.4780 | 0.4630 | 0.2479 | 0.4890 | 0.5108 |
| | $\Delta acc_w$ | 0.8288 | **0.4653** | 0.4740 | 0.6971 | 0.2799 | **0.2400** |
| APGD-CE | acc | 0.1338 | 0.3095 | 0.3142 | 0.2152 | 0.2197 | 0.2208 |
| | $\Delta acc$ | 0.8579 | 0.6384 | **0.6196** | 0.7360 | 0.5754 | **0.5658** |
| | $acc_w$ | 0.1580 | 0.4396 | 0.4089 | 0.2745 | 0.4756 | 0.4833 |
| | $\Delta acc_w$ | 0.8339 | **0.5082** | 0.5355 | 0.6647 | 0.2995 | **0.2809** |
| APGD-T | acc | 0.3931 | 0.4085 | 0.3944 | 0.2618 | 0.2235 | 0.2182 |
| | $\Delta acc$ | 0.5826 | 0.5227 | **0.5225** | 0.6618 | **0.5680** | 0.5709 |
| | $acc_w$ | 0.4985 | 0.5663 | 0.5295 | 0.5079 | 0.4827 | 0.4818 |
| | $\Delta acc_w$ | 0.4758 | **0.3665** | 0.3984 | 0.3796 | 0.2892 | **0.2832** |
| FAB-T | acc | 0.9259 | 0.8389 | 0.8116 | 0.7596 | 0.5109 | 0.5001 |
| | $\Delta acc$ | **0.0168** | 0.0199 | 0.0174 | 0.0187 | **0.0126** | 0.0165 |
| | $acc_w$ | 0.9376 | 0.8813 | 0.8683 | 0.8075 | 0.6746 | 0.6669 |
| | $\Delta acc_w$ | 0.0140 | 0.0141 | **0.0136** | 0.0136 | **0.0065** | 0.0079 |
| SQUARE | acc | 0.8062 | 0.7443 | 0.7044 | 0.6050 | 0.4101 | 0.3847 |
| | $\Delta acc$ | 0.1439 | **0.1304** | 0.1472 | 0.2184 | **0.2074** | 0.2435 |
| | $acc_w$ | 0.8389 | 0.8100 | 0.7696 | 0.6961 | 0.6061 | 0.5516 |
| | $\Delta acc_w$ | 0.1178 | **0.0939** | 0.1497 | 0.1289 | **0.1074** | 0.1794 |

fication like random crop, rotation, width and height shift. The clean accuracy for Resnet50 is nearly comparable to some of the recently reported approaches, such as Rebuffi et al. (2021b).

## 3.2 ATTACKS INVESTIGATED

Our proposed tessellated mechanism can be applied to any base network. As particular choices of the base network, we experimented with the popular image classification architectures, namely the ResNet50 He et al. (2016).

For our experiments, the black-box untargeted versions of the two most standard adversarial attacks - FGSM (Fast Gradient Signed Method (Goodfellow et al., 2015)) and PGD (Projected Gradient Descent (Madry et al., 2018)), were applied on the neural models, i.e., both on the base networks and their tessellated versions. Additionally, we conducted experiments with more recently proposed attacks, namely APGD-CE (Auto Projected Gradient Descent Attack with Cross Entropy Loss) (Francesco Croce, 2020) and APGD-T (Targeted Auto Projected Gradient Descent Attack), FAB-T (Targeted Fast Adaptive Boundary Attack) (Croce & Hein, 2019), and Square Attack, which is a query-efficient black-box adversarial attack (Andriushchenko et al., 2019).

For consistency, all the adversarial examples were generated apriori, and were used across all our experiments. The FGSM examples were generated by applying $\ell_\infty$ perturbations of $\varepsilon = 8/255 = 0.03$, on normalized test sets, which is a standard practice adopted in existing research Goodfellow et al. (2015); Madry et al. (2018). Similarly, to generate the PGD examples, we used $\varepsilon = 0.03$, and $step\_size = 2.5 \times \varepsilon / num\_steps$, where $num\_steps$ is the number of iterations of the PGD algorithm, the value of which was set to 20 as per Madry et al. (2018). All the more recent attacks were generated by applying $\ell_\infty$ perturbations of $\varepsilon = 8/255 = 0.03$, on normalized test sets.

In all the attacks we assume a black-box scenario with the proxy victim network to be a standard ResNet50.

Since the objective of our study is rather to demonstrate the robustness of the tessellated networks over equivalent non-tessellated ones, we do not study defence mechanisms based on adversarial retraining. Moreover, an important point to note in our experiment setup is that we investigate a black-box adversarial situation, i.e. where the adversary has no information about the model architecture, parameters or the gradients. We used the adversarial attack methodologies, e.g., PGD, FAB-T etc., to perturb the CIFAR-10/100 images using the base model (in our case, either Resnet or the WideResNet). These perturbed images are then used as the test set to evaluate the defence mechanisms. This is in contrast to the experiment settings used in the RobustBenchmark framework Croce et al. (2020), where instead of leveraging information of the base network (as in our case), the attack mechanisms make a direct use of information from the original models. As such, the results obtained on our proposed architecture are not directly comparable with those reported in the RobustBenchmark.

## 3.3 EVALUATION METRICS

The objective of the experiments is to measure how effective is a defence mechanism. Since different base networks (on which tessellation is applied as a defence mechanism) when trained on unperturbed data samples will likely yield different accuracy values, for a fair comparison of the robustness measure across different attacks, one should compute the decrease in the accuracy of the attacked network (trained with adversarial samples) relative to the clean (unperturbed) case. In addition to reporting these relative accuracy values, we also report the individual accuracy values of the base and the tessellated networks prior to and post adversarial attacks, denoted respectively as $\mathrm{acc}$ and $\mathrm{acc}_A$. All our experiments report results with a 5-fold cross validation.

In addition to reporting the above metrics, we also evaluate the effectiveness of our proposed method with a metric that incorporates the uncertainties of the predictions. The closer the softmax probability distribution is to the uniform, the higher is the uncertainty in predictions, which is not a desirable criteria. We intend to capture the effectiveness and the confidence of predictions with a single metric, which we call the *weighted accuracy*. More precisely, instead of simply employing the $\mathrm{argmax}$ on the output softmax to predict the correct class, we multiply the maximum component of the softmax probability with the binary indicator of whether the prediction is correct. We then report the average of these values computed over all instances in the evaluation set, and denote it by $\mathrm{acc}_w$.

A neural model that exhibits a larger absolute value of weighted accuracy ($\mathrm{acc}_w$) after an attack would indicate more resilience against the attack. Similar to the unweighted case, we also report the relative changes in the weighted accuracy values as well. We denote these relative changes in accuracy and weighted accuracy values as $\Delta\mathrm{acc}$ and $\Delta\mathrm{acc}_w$ respectively, and they are computed as

$$\Delta\mathrm{acc} = \frac{\mathrm{acc} - \mathrm{acc}^{attack}}{\mathrm{acc}}, \quad \Delta\mathrm{acc_w} = \frac{\mathrm{acc_w} - \mathrm{acc_w}^{attack}}{\mathrm{acc_w}} \tag{1}$$

## 3.4 TESSELLATION AND NETWORK PARAMETERS

We experimented with different configurations of the tessellated ResNet50. In particular, we experimented with different tiling schemes - regular or aperiodic, and also with different number of tiles (the value of $k$ as described in Section 2.2). The objective of applying a number of different attacking mechanisms (as described in Section 3.2) is to demonstrate that our proposed method, being attacker method agnostic, should provide a robust defence against each.

For regular tessellation a total of $k = 16$ equal sized squares were used both for the CIFAR-10 and CIFAR-100 $32 \times 32$ images. For aperiodic tessellation, we split the input images into $k = 20$ sub-rectangles using the split/merge generation process described in Section 2.2.2. Owing to the random initialization of the aperiodic spiral pattern generation process, multiple tessellations with identical values of $k$ is likely to be generated. We report the average value of the evaluation parameters over 5 such configurations. The architectural details are provided in the appendix (Section A.

We use the categorical cross-entropy as the loss function. We train with SGD and Adam optimizer using a batch size of 128 and 100, respectively for ResNet and Wide-ResNet. All the models were trained for 100 epochs.

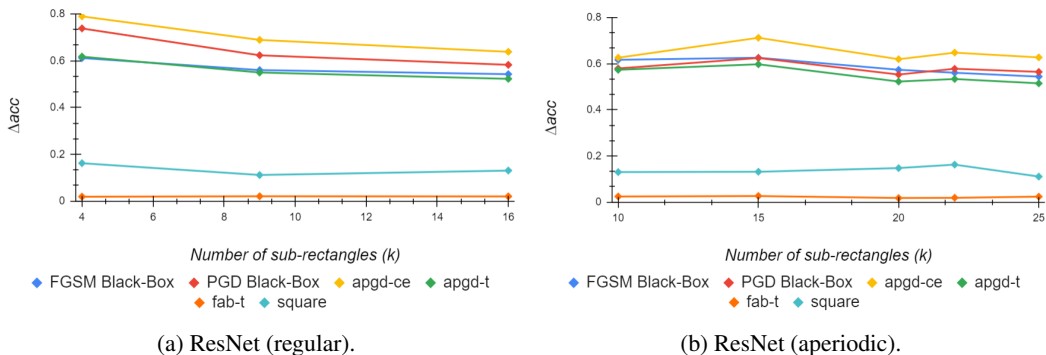

(a) ResNet (regular).                    (b) ResNet (aperiodic).

Figure 3: Sensitivity of relative drops in accuracy values ($\Delta$acc) with variations in the number of tiles ($k$) for several attacks on the CIFAR-10 dataset. Similar to the results of Table 2, each attack used a perturbation of $\ell_\infty, \varepsilon = 0.03$.

Table 3: Effectiveness of Wide-ResNet-16-8 (WRN) and regular ($k = 16$) Tessellated Wide-ResNet-16-8 (TWRN-RG) against $\ell_\infty$-based FGSM and PGD adversarial attacks on the PlantVillage dataset ($\varepsilon = 8/255$).

| Attack | Metric | WRN | TWRN-RG |
|--------|--------|------|---------|
| NONE | acc | 0.9618 | 0.9232 |
| | $\mathrm{acc_w}$ | 0.9745 | 0.9388 |
| FGSM | acc | 0.3459 | 0.5815 |
| | $\Delta$acc | 0.6404 | 0.3702 |
| | $\mathrm{acc_w}$ | 0.5876 | 0.6668 |
| | $\Delta\mathrm{acc_w}$ | 0.3970 | 0.2897 |
| PGD | acc | 0.3168 | 0.7525 |
| | $\Delta$acc | 0.6706 | 0.1849 |
| | $\mathrm{acc_w}$ | 0.4928 | 0.8051 |
| | $\Delta\mathrm{acc_w}$ | 0.4943 | 0.1424 |

## 4  RESULTS

**Main Observations.**   Table 2 displays a comparison between the base network Resnet50 and its tessellated version obtained with optimal settings for the number of tiles. The following observations can be made. *First*, tessellated architectures when trained with clean data (i.e., no attack) result in a lower prediction effectiveness, e.g., compare the acc and $\mathrm{acc_w}$ values for ResNet (RN) and tessellated ResNet (TRN). *Second*, for a number of different attacks, it can be seen that the post-attack accuracy of tessellated networks is mostly higher than its base counterpart, e.g., compare the acc value of a post-PGD attack on ResNet (RN) with that of a tessellated one (the corresponding TRN column values in the same row).

The *third*, and perhaps the most important, observation is that the relative drops in accuracy and weighted accuracy values (Equation 1) are lower in tessellated versions of the base networks for almost all the attacks. *Fourth*, we observe that a Aperiodic tessellated network mostly outperforms its regular counterpart (see that most bold-faced values occur along the 'Aprd' columns).

We observe that the drops in classification accuracy ($\Delta$acc) is smaller for the tessellated ResNet network as compared to basic ResNet network. We further observed that the aperiodic tiling method seem to have lower accuracy drop than regular tessellation in most cases.

**Experiments on larger images.**   In our next experiment, we investigate the effectiveness of our proposed defence approach on images much larger than the ones in CIFAR datasets. For these experiments, we focus only on the best performing model (as per the results of Table 2), namely the tessellated ResNet50. Trends similar to that of Table 2 is observed in Table Table 3. It can be

seen that even on larger images, the tessellated network is significantly more robust compared to the non-tessellated ones for both FGSM and PGD attacks.

**Sensitivity analysis.** Since the number of sub-rectangles ($k$) used in a tessellation is a hyper-parameter of our model, we now investigate how varying this hyper-parameter (i.e., different tessellation configurations) can affect the robustness against several attacks.

From Figure 3, it can be seen that the resilience to attacks is lower when a very small number of rectangles are used for configuring a tessellated network; note that the $\Delta(\mathrm{acc})$ values are higher for small values of $k$. Similarly, the robustness also decreases (i.e., acc values increase) when a large number of tiles is used.

## 5 CONCLUSIONS AND FUTURE WORK

We presented a novel defence mechanism that transforms any base neural model for image classification to a tessellated network. An input image is partitioned into a number of non-overlapping sub-rectangles. Each sub-rectangle is then processed by a dependent branch of a base neural model (e.g., the ResNet) terminating in dense layers. The output of these branches are concatenated and passed through another dense layer to obtain the final softmax classification scores. The key idea behind this approach is that none of the computational branches have complete information on the overall input image, as a result of which it becomes difficult for an adversary to craft adversarial examples by leverage the gradient information.

We investigated two different tiling strategies - regular and aperiodic, and experimented with the base networks, namely ResNet50. We empirically demonstrate that standard non-tessellated networks are more vulnerable to adversarial attacks, whereas the tessellated ones are more robust against these attacks.

Studies on other space partitioning techniques may help in devising more diverse classes of tessellated networks. Similarly, tessellation structures for other networks such as Efficient nets (Tan & Le, 2019) and Dense nets (Huang et al., 2017) may be considered in future.

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

## A  APPENDIX

In this section, we provide the architecture details of two sample configurations for tessellated ResNet-50 employed for the CIFAR-10 and CIFAR-100 image classification (the schematic diagram is shown in Figure 1). This enlisted configurations use $k = 20$ tiles for Aperiodic tessellation and $k = 16$ for periodic tessellation since this value yielded the optimal results (we report the sensitivity later in Figure 3).

**Aperiodic tessellated ResNet-50 (with** 20 **sub-rectangles)**: Crop-layers (all parallel custom crops); 4X4 Upsampling and Resizing Layer convert to 224*224; Pretrained ResNet50; GlobalAveragePooling; Flatten; Dense 256; Concat; Dense 128; Dense 10.

**periodic tessellated ResNet-50 (with** 20 **sub-rectangles)**: Crop-layers (uniform 8*8); 4X4 Upsampling and Resizing Layer convert to 224*224; Pretrained ResNet50; GlobalAveragePooling; Flatten; Dense 256; Concat; Dense 128; Dense 10.

