# OpenReview forum: "Tessellated Neural Networks: A Robust Defence against Adversarial Attacks"
_ICLR.cc/2023/Conference — Submitted to ICLR 2023_

### Official Review · Reviewer_n7Ya · 2022-10-21

**Confidence:** 4
**Correctness:** 2
**Technical Novelty And Significance:** 4
**Empirical Novelty And Significance:** 2
**Recommendation:** 3

**Clarity, Quality, Novelty And Reproducibility:**

The work is novel and original. The writing is not clear. The quality is not so good.

**Strength And Weaknesses:**

Strength: The idea of design a new architecture that allows the model to do implicit feature transformation to defend against adversarial attack  is novel.

Weakness:
1. The writing is poor. There are seven paragraphs in the Introduction section that focus on the background information of adversarial attack, and only four paragraphs talking about the proposed method. Many information in the first seven paragraphs seem irrelevant to the main idea in this paper. Reading the introduction is boring and tedious. Also, I fail to find any description regarding to the evaluation behavior of tesellating input images. It seems to me the two non-uniform tessellation methods are random operations, and how to keep the model predictions deterministic during evaluation is still a mystery based on this paper. Plus, does all the patches share the same model weight? Or there is a separate branch for each one of them? In the latter case, the model's computational cost is multipled.
2. Lack of citations. This paper makes claims without any citation. For example, the last sentence in the first paragraph 'Architectural changes in the network topology is a promising means of achieving adversarial robustness ' is one without any citation.
3. Lack of (experimental) proofs. The motivation of this paper is a hypothesis: 'modification of the network structure leading to implicit feature transformation, cropping, masking, and distillation may result in improved robustness'. To verify this hypothesis, the authors should at least show some experiments demonstrating that explicit image-level cropping or masking or other transformations will help the model defend against adversarial attack, and embedding such a prior knowledge into the architecture design is a similar but more efficient implementation. However, I fail to see any such experiment.
4. Poor experiment designs. This paper introduces a new metric called weighted accuracy (accw) without enough justification. This paper only experiments on very simple CNNs. The results in Table 2 show that a modern architecture like ResNet can easily beat the proposed T-2D-CNN in terms of both clean accuracy and robustness, let alone lots of more powerful networks have been developed after ResNet. Furthermore, the images in the two used datasets in this paper are of small resolutions (28 or 32). What's the effect of the proposed T-2D-CNN on larger images, like 224x224?

**Summary Of The Paper:**

This paper tries to improve adversarial robustness via a new perspective: tessellated 2D convolutional network as a divide and conquer defence. The input image is first divided into several non-overlapping patches (regular or irregular), sent to parallel branches, and then the features from different patches are aggregated inside the network. The underlying idea is that such a network architecture leads to implicit feature transformation, cropping, masking and thus improve model robustness. Results on Fashion-MNIST and Cifar-10 validate the effectiveness of the proposed model to some extent.

**Summary Of The Review:**

In general, I think this work proposes a interesting idea, but fail to prove its effectiveness. There is still a lot to improve. Maybe this work can be published at another conference after polishing.

---

### Official Review · Reviewer_KrRX · 2022-10-23

**Confidence:** 5
**Correctness:** 2
**Technical Novelty And Significance:** 4
**Empirical Novelty And Significance:** 2
**Recommendation:** 3

**Clarity, Quality, Novelty And Reproducibility:**

The paper proposes a novel method. The method is well motavated the clearly explained. The quality of the paper can be improved with more solid experimental settings (see weakness for detail).

**Strength And Weaknesses:**

## Strength
1. The paper proposes a novel model architecture that utilize diverse partition of the input image. To my knowledge this haven't been explored in previous work
2. The paper is overall well written and easy to follow
3. The proposed method shows natural improvement on blackbox robustness over regular model without the need of adversarial training

## Weakness
1. The experiment provided is inadequate to fully support the claim on improved blackbox robustness. All attacks provided in the paper are generated with a regular ResNet model, which naturally will have better transferability on a regular model than the proposed model. To fully verify the robustness of the proposed architecture experiment should be conducted on transfer attack generated against an independently trained TRN model with the same partition configuration. This is also a valid threat under blackbox setting, as only model architecture is known by the attacker.
2. The proposed method is only tested against naive model, without any other baseline blackbox robustness improvement methods
3. The attack effectiveness in Table 2 is doubtful. Even on the regular model the attack is not effective, especially for the FAB-T and SQUARE attack, which is unexpected. More information is needed on how the attack is implemented, an how is the convergence behavior in the generation of the attacks
4. The paper mainly uses the $\Delta$acc as the metric, which does not make sense. Given the tradeoff between clean accuracy and robustness, we want the robust model to have a better accuracy under attack than the clean model, not just a smaller accuracy drop. Otherwise a randome guess model will always have the best $\Delta$acc = 0 under no matter what attack

**Summary Of The Paper:**

This paper provides a novel CNN architecture that process input images as non-overlapping regions or tiles. The paper claims that the partition reduces the input dimension and diversifies the model architecture, which may provide effective defense against adversarial attack.

**Summary Of The Review:**

In summary, the paper propose an novel and intereting model architecture against blackbox adversarial attack. However, the experiment results are inadequate to support the claim that the proposed method is effective. Thus I would recommend rejection for now.

---

### Official Review · Reviewer_nZEg · 2022-10-24

**Confidence:** 4
**Correctness:** 3
**Technical Novelty And Significance:** 2
**Empirical Novelty And Significance:** 2
**Recommendation:** 3

**Clarity, Quality, Novelty And Reproducibility:**

The paper is well-written but lacks novelty to make sufficient contributions to the research community.

**Strength And Weaknesses:**

Strength:

1. This paper is well-written and easy to follow.

2. The proposed method to tessellate neural networks is simple and clear.

Weakness:

1. The proposed method suffers too much from in-distribution accuracy drop. Based on the results shown in Table 2, the proposed new method significantly reduces classification accuracy on the clean images. For example, on CIFAR-100, the standard ResNet can achieve around 77% accuracy whereas the proposed method can only achieve 50%. The significant in-distribution accuracy drop makes this method less practical. In addition, the low in-distribution accuracy makes the evaluation metric: difference between clean accuracy and adversarial accuracy is less reasonable.

2. The proposed method does not have inspiring motivations and lack of sufficient contributions to the robustness research community. For example, in the Section 2.2.2, the authors discussed "Why should tessellated networks be robust against adversarial attacks" and mentioned that "this offers an implicit ensemble". Since ensemble does not harm in-distribution accuracy, I am not fully convinced by using this new training pipeline instead of simple ensemble if they share similar motivations.


**Summary Of The Paper:**

This work propose a training pipeline to help defend against adversarial robustness. Specifically, an input image is partitioned into a number of non-overlapping sub-rectangles. Then each sub-rectangle is process by a shared branch of base neural network and further merged together to make a prediction. They show that this new training pipeline suffers less from adversarial attacks.

**Summary Of The Review:**

See above.

---

### Official Review · Reviewer_z2SF · 2022-10-25

**Confidence:** 3
**Clarity, Quality, Novelty And Reproducibility:** The paper is clear. In this regard, t…
**Correctness:** 3
**Technical Novelty And Significance:** 2
**Empirical Novelty And Significance:** 2
**Recommendation:** 5

**Strength And Weaknesses:**

Strength

1. The paper is clear, and I had no trouble understanding it.
2. The idea of this paper is simple and intuitive.

Weaknesses

1. There are no results on white-box attacks which are generated from tessellated network. The results on black-box attacks only show that this approach reduces the transferability of black-box adversarial attacks, rather than improving the adversarial robustness. Moreover, this is not consistent with your proposal that the ensemble of diverse network structures have been shown to exhibit robustness.
2. This paper lacks more theoretical analysis of the tessellated network.
3. This paper lacks related work on tessellation in images and in NN.


**Summary Of The Paper:**

This paper proposes a novel divide-and-conquer based approach of tessellating a base network architecture. Each tessellated network involves a separate set of learnable parameters. Experiments show this method is more robust against black-box attacks.

**Summary Of The Review:**

This work presents an architecture with simple and intuitive idea, but there are no results on white-box attacks which are generated from tessellated network. The contributions to adversarial robustness are only marginally significant.

---

### Decision · Program_Chairs · 2023-01-20

**Decision:**

Reject

**Justification For Why Not Higher Score:**

The authors of this paper did not provide a rebuttal to address the concerns raised by reviewers.

**Justification For Why Not Lower Score:**

N/A

**Metareview: Summary, Strengths And Weaknesses:**

This paper presents a novel neural architecture for defending against adversarial attacks. While reviewers found the content of the paper to be interesting, several major concerns were raised, such as the lack of evaluations against white-box attacks and strong transfer attacks, the significant drop in in-distribution accuracy, and the lack of motivation or theoretical analysis. Unfortunately, the authors have not provided a rebuttal addressing these concerns.

The AC encourages the authors to carefully tackle these concerns and make a stronger submission next time.